# What Do Young Adults Think about the Safety of Over-the-Counter Analgesics? Findings from a Cross-Sectional Survey

**DOI:** 10.3390/pharmacy9010054

**Published:** 2021-03-05

**Authors:** Tahlia Duyster, Sara S. McMillan, Ella Whately, Fiona S. Kelly

**Affiliations:** 1School of Pharmacy and Pharmacology, Gold Coast Campus, Griffith University, Southport 4215, Australia; tahlia.duyster@health.qld.gov.au (T.D.); s.mcmillan@griffith.edu.au (S.S.M.); e.whately@griffith.edu.au (E.W.); 2Gold Coast Hospital and Health Service, Southport 4215, Australia

**Keywords:** analgesics, attitudes, non-prescription, risk, safety, young adults, over the counter

## Abstract

Analgesics are commonly used over-the-counter (OTC) medicines readily available for purchase, sometimes without advice of a health professional. However, analgesics can cause harm even when taken according to dosing recommendations. Young adults may be more vulnerable to harm if they perceive low risk with OTC analgesic use, or struggle to interpret dosing instructions. This study aimed to explore factors affecting how young adults use OTC analgesics and associated perceptions of safety. An online survey was distributed to school-leavers and university students (17 to 25 years), in South-East Queensland, Australia, in the period November–December 2019. Most of the 302 respondents (school-leavers *n* = 147, university students *n* = 155) did not use analgesics frequently. School-leavers deferred to parents for analgesic information, while university students preferred the internet. The majority of respondents appeared safety conscious and did not take outside indicated use or instructions. However, a small proportion reported taking analgesics for an inappropriate indication. The difference in preferred source of analgesic information may reflect experience with analgesic use, increasing autonomy or decreased parental influence. Whilst it is encouraging that the majority of young adults appeared safety conscious, greater insight is needed into factors influencing decision making on OTC use, e.g., medicines knowledge, and changes with increasing age.

## 1. Introduction

Self-administration of over-the-counter (OTC) medications is common; in 2018, Australians spent an estimated $5.4 billion dollars on OTC medications [1], with most adults using at least one medication per month, most commonly analgesics [2]. Examples of OTC analgesics include paracetamol and non-steroidal anti-inflammatory drugs (NSAIDs) including aspirin, ibuprofen, naproxen, diclofenac and mefenamic acid. The analgesics aspirin, paracetamol and ibuprofen can be purchased from supermarkets or petrol stations in small quantities without health professional advice as unscheduled (general sales list) medications. Larger quantities or analgesics deemed to require health advice with purchase are restricted to pharmacy sale (Pharmacy Medicines), and analgesics associated with a higher risk of harm or misuse are supplied by the pharmacist (Pharmacist-Only Medicines) [3,4].

Frequency of use and ease of procurement engender a perception of OTC analgesics as relatively harmless, despite reported harm [5,6]. NSAIDs can cause severe gastrointestinal ulceration, bleeding, and renal and cardiac complications even when taken as recommended [7,8]. Paracetamol toxicity is the most prevalent cause of acute liver failure in the Western world when taken inappropriately or in excess [9,10,11,12]. In Australia, paracetamol is the medication most frequently used in overdoses, and was associated with a 108% increase in paracetamol-related acute liver toxicity between 2007 and 2017 [12]. Additionally, paracetamol has also been associated with a number of serious hypersensitivity reactions, such as angioedema, Steven–Johnsons syndrome, toxic epidermal necrolysis and anaphylaxis [13]. Internationally, strategies implemented to minimise paracetamol-related harm include restricted pack size and location of sale, resulting in reduced paracetamol-related calls to poisons centres [14]. Young adults appear to be at high risk of paracetamol-related harm; in Australia in 2017, there were 4577 cases of paracetamol toxicity recorded with the National Hospital Morbidity Database involving young adults aged 15–24 [12].

As young adults (typically aged 15 to 25 [15]) become accountable for their health, they make autonomous decisions about procuring and taking medication [5]. Inappropriate use of OTC medications and subsequent risk of harm could increase if health-related decisions are uninformed [9], young adults engage in risk-taking behaviours [16], or if they perceive less risk for OTC medications than older adults [6]. Young adults reportedly believe that they are immune to adverse effects because they are young and healthy [5], and studies have identified that this population can struggle to identify active ingredients and incorrectly interpret dosing instructions [9,10,11,17]. For example, American adolescents were unaware of what paracetamol was, despite having taken it recently [11], and Swedish youth believed that paracetamol was additionally used for sedating, anxiolytic and performance-enhancing properties [5]. Young adults may be less familiar with NSAID analgesics; nearly two-thirds of American high-school students (61.8%) were unaware if ibuprofen and naproxen could be taken together [18].

Many factors appear to influence a young adult’s choice of medication, including the media [19], parental influence [9,20], accessibility of medicines and related health information [21,22], and medication taste, coating, and packaging [5]. Access to health information was of low priority for young adults in comparison to older adults [21]. Young adults appeared less likely to use OTC medication if their parents had advised them of the associated risks [9], and more likely to use OTC analgesics appropriately if their parents were health professionals [20]. Selected studies report that attitudes towards, and preferred sources of information about, OTC analgesics are primarily influenced by family [5,23,24,25], yet other studies found pharmacists [5,20,26,27], doctors [18,28], or the internet [17], as preferred source of information. Young adults infrequently consult peers about medication [18,20,23,25,26], except for females seeking dysmenorrhea-related advice [5]. However, peers may influence behaviour and potentially encourage risky analgesic-taking behaviour [9]. In one study, young adults described peer influence to take analgesics despite not having symptoms [5].

Even when appropriate information is available, if young adults have poor health literacy, this could lead to unsafe dosing, inappropriate medication selection, and the use of multiple medications without health professional advice to guide decisions [10,20,29,30]. Health literacy is defined as ‘the degree to which individuals have the capacity to obtain, process, and understand basic health information and services needed to make appropriate health decisions’, and is a key factor in empowering one’s own health [31,32]. Health literacy involves three main domains: functional, interactive, and critical literacy [20,33]. Functional literacy encompasses basic reading and writing skills, for example finding information on medication packaging. Interactive literacy explores cognitive ability and social skills to find and use information, such as communication with health professionals [33]. Critical literacy reflects advanced application and analysis of health information to improve personal and community health [20,33]. Studies have shown that young adults may have adequate functional literacy, but limited critical and interactive literacy [18,20], which could contribute to poor knowledge and application of skills in regard to analgesic use.

Internationally, the medication-taking behaviour of young adults has been influenced by various factors, including familial attitudes, sources of information, medication knowledge, health literacy, peer influence, and perceptions of safety [10,17,20,24,28,29]. However, to our knowledge no studies have explored OTC analgesic use by Australian youth or identified factors that influence this. This study aimed to explore the factors influencing how young Australian adults use OTC analgesics, and their perceptions of associated risks.

## 2. Materials and Methods

### 2.1. Study Population

A prospective, cross-sectional survey was undertaken in young Australian adults aged 17 to 25 years. A sample size of 250 participants was estimated based on advice from a statistician and informed by previous research [34].

Two recruitment strategies were utilised. School-leavers were recruited in person at the Safer Schoolies Response, Gold Coast Schoolies Festival on the 16th of November 2019, an event that attracts approximately 40,000 school-leavers [35,36]. Two researchers (TD and FK) invited people appearing to be within the age range or wearing the official Schoolies lanyard or wristband to complete the ten-minute survey using iPads. School-leavers were approached sequentially with the researchers each using four iPads to optimise recruitment. The option of scanning a survey quick response (QR) code was also available so that participants could complete the survey on their phone. Two researchers were present until 2:00 pm and it is estimated that 3005 school-leavers checked into the wristband centre during this time.

The second recruitment process occurred at one Queensland University between November and December 2019. A multimodal approach was adopted to engage university students, including English-language posters displayed in approved locations, study promotion in a monthly research project email to all enrolled students, directly approaching students on campus, and in-person promotion at three classes at one school within the health field, each comprised of approximately thirty students. Calculation of response rate was limited by multiple recruitment methods.

### 2.2. Survey Development

A 17-item survey (Appendix A) was developed comprising six topic areas: (i) demographics; (ii) sources of analgesic information; (iii) safety and risk perception, including exceeding maximum doses and non-indicated use; (iv) use of analgesics and factors affecting this; (v) management of pain symptoms; and, (vi) preferred location to purchase OTC analgesics. Questions were informed by the literature or adapted from previously published surveys. In the final survey, one question was adapted from each of four studies including Kelly et al. [18], Shone et al. [10], Holmström et al. [5], and Klimaszova et al. [17]. Although some of these were validated surveys, these instruments did not meet all aims of this particular study, and thus selected questions were adapted to address the specified objectives. The total number of questions asked of everyone was 17, with minor variations for school-leavers and university students; three questions had additional branching logic depending on participant response.

The survey was piloted by three registered pharmacists and 18 young adults known to the research team to assess face and content validity [37]. Nine questions were subsequently amended; for example adding in an optional ‘explain’ box for Likert-scale questions on safety.

A web-based survey was created via the online survey platform, SurveyMonkey^®^, as young adults engage more with web-based surveys in comparison to paper-based surveys [38,39,40]. The number of survey questions varied based on participant responses through logic branching. For example, if the respondent replied ‘yes’ to the question: ‘*I have taken an over-the-counter pain medicine for a reason other than pain*’, then an additional question was asked: ‘*what pain medicine and for what reason?’*

Strategies to improve participation rates included limiteing survey completion time to 10 min, and a $5 food voucher incentive for school-leavers or the opportunity to win a small gift for university students on survey completion [41].

Ethics approval was obtained from the University Human Research Ethics Committee (2019/915). Parental consent was not required for participants aged 17 years as there were no significant risks or research burden for participants. Survey completion was accepted as informed consent. Written consumer information about analgesics normally available from pharmacies was offered to participants on survey completion.

### 2.3. Data Analysis

Data were exported directly into a Microsoft^®^ Excel^®^ workbook [42] and statistical analysis software (SPSS^®^ version 25) [43]. Data were cleaned and additional variables added to explore the potential influence of factors such as rurality of location and access to health advice via a pharmacy. Postcodes were converted to the Pharmacy Access/Remoteness Index of Australia (PhARIA) code using the 2019–2020 PhARIA criteria [44]. This code determines degree of remoteness in terms of access to a pharmacy categorised from 1 (highly accessible) to 6 (very remote). Where a postcode had multiple PhARIA codes, the code reflecting greater access to a pharmacy was chosen. Selected variables, such as the five-point Likert-scale question, namely Agree Strongly, Agree Somewhat, Neutral, Disagree Somewhat, Disagree Strongly, or I Don’t Know, were recoded to Agree, Disagree, Neutral, or I Don’t Know options. In addition, frequency of analgesic use was recorded from the original options: Never, Rarely, Few times a year, Every week, Every month or Every day, into Never/Rarely, Few times a year, Monthly, or Daily/Weekly. These selected variables were collapsed when multiple factors were included in analysis for statistical purposes. Descriptive and inferential statistics were used for data analysis; chi-square tests were used to test for differences in proportions between groups [45]. A p-value of less than 0.05 was considered significant.

## 3. Results

### 3.1. Demographics

The survey was completed by 313 young adults; 11 incomplete responses were omitted, leaving 302 usable responses. Respondents had an average age of 19.1 years (SD = 2.5 years), with an average age of 17.2 years (SD = 0.4 years) for school-leavers and 20.9 years (SD = 2.2 years) for university students (Table 1). Participants were mostly from Queensland (*n* = 278/302, 92.1%), 99.3% (299/301) in PhARIA 1 locations, and two-thirds were female (*n* = 196, 64.9%). Only eight participants self-identified as being of Aboriginal or Torres Strait Islander descent. Almost one-fifth of participants (*n* = 56/299, 18.7%) self-reported a chronic health condition, 41 of these were university students. Almost one-quarter of the respondents (*n* = 68/299, 22.7%) had at least one parent working in a health-related field, and 39.8% (*n* = 102/256) were intending to (23 school-leavers) or currently studying (79 university students) a health-related degree.

### 3.2. OTC Analgesic Use

Approximately half of the young adults surveyed reported rare (*n* = 81/302, 26.8%), or infrequent use, i.e., a few times a year (*n* = 69/302, 22.8%) of analgesics. Age did not significantly influence frequency of use when 17 year old school leavers were compared with older university students (*p* = 0.9, *df* = 3, $\chi^{2}$ = 0.5).

Females were significantly more likely to report frequent use of analgesics than males (*p* < 0.001, *df* = 3, $\chi^{2}$ = 32.7). For example, 43 of the 50 respondents using analgesics on a daily or weekly basis, and 54 of the 68 respondents using analgesics monthly, were female. Significantly more respondents with a chronic health condition used analgesics on a frequent basis in comparison to young adults without a chronic condition (*p* < 0.001, *df* = 3, $\chi^{2}$ = 28.7). Twenty-one of the 53 young adults with a chronic health condition used analgesics weekly or daily.

There were no significant differences in analgesic use between students currently or prospectively studying health and non-health-related fields (*p* = 0.9, *df* = 3, $\chi^{2}$ = 0.8), or young adults with parents working or not working in health care (*p* = 0.6, *df* = 3, $\chi^{2}$ = 1.9).

#### 3.2.1. Location of Purchase

When asked to indicate where OTC analgesics were purchased, the majority of young adults (*n* = 230/302, 76.2%) obtained these from pharmacies, followed by supermarkets (*n* = 154/302, 51.0%) and petrol stations (*n* = 12/302, 4.0%). These findings were similar between school-leavers and university students.

#### 3.2.2. Factors Influencing Decision Making

The most common factors identified by young adults as influencing their decision to use analgesics were previous experience with the medication (school-leavers: *n* = 65/147, 44.2%; university students: *n* = 115/155, 74.2%), cost (school-leavers: *n* = 43/147, 29.3%; university students: *n* = 104/155, 67.1%), and medication accessibility (school-leavers: *n* = 33/147, 22.4%; university students: *n* = 49/155, 31.6%) (Table 2). Medication packaging was the option selected least often by school-leavers (*n* = 14) and university students (*n* = 15).

Fifteen university students selected ‘other’ factors that influenced their decision to use analgesics, and these included strengths of analgesic, suitability for the type of pain, active ingredient, side-effect profile, comorbidities, and formulations that could be chewed.

#### 3.2.3. Management of Pain Symptoms

Nearly half of the survey respondents reported using OTC analgesics at the first sign of pain (*n* = 127/288, 44.1%), with significantly more school-leavers (*n* = 80/127) than university students (*n* = 47/127) (*p* < 0.001, *df* = 4, $\chi^{2}$= 32.8). Nearly twice as many university students selected the option of treating the cause of the pain first before self-medicating with analgesics (*n* = 77/116) compared to school-leavers (*n* = 39/116). Some university students would try and ‘tough it out’ until pain either went away or became intolerable. A smaller proportion of young adults (*n* = 17/288, 5.9%), mostly university students (*n* = 12), reported using alternative treatments first, such as herbal remedies.

### 3.3. Source of Information

When study participants ranked their preferred source of OTC analgesic information (1 = most likely to 6 = least likely), the majority of school-leavers placed parents as their most preferred source of information (*n* = 78/142, 54.9%,), whereas university students preferred the internet (*n* = 47/150, 31.3%) (Figure 1). Doctors (School-leavers: *n* = 84/138, 60.9%; University students: *n* = 92/150, 61.3%) and pharmacies (School-leavers: *n* = 69/137, 50.4%; University students: *n* = 108/149, 72.5%) featured in the top three preferences for both groups. Young adults in this study rarely rated their friends as a preferred source of information about analgesics (*n* = 17/287).

### 3.4. Safety Perception of OTC Analgesics

Two-thirds of all young adults surveyed (*n* = 194/298, 65.1%) disagreed that: ‘*Taking more medicine than stated on the pack gives a better effect’*(Table 3). Twenty-four of the 37 participants who agreed with this (12.4%) were university students and nine were enrolled in a health science degree.

Altogether, more young adults disagreed (*n* = 125/296, 42.2%) with the statement, ‘*I think over-the-counter pain medicines are safe (they cannot hurt me)’,* than agreed (*n* = 91, 30.7%), yet approximately one-quarter remained neutral or did not know. A significantly greater number of university students disagreed with this statement (*n* = 80/125, 64.0%) compared to a more mixed response from school-leavers (*p* < 0.001, *df* = 3, $\chi^{2}$ = 19.3). Similar trends in disagreement were observed ireespective of gender, whether students were working or studying in a health-related field, had a parent working in health care, or a concurrent health condition.

A smaller number of school-leavers reported that they would share, or have shared, OTC analgesics with their friends (*n* = 51/172, 29.7%), than university students (*n* = 121/172, 70.3%) (*p* < 0.001, *df* = 3, $\chi^{2}$= 58.4).

Seventy of 291 respondents (38 school-leavers and 32 university students) indicated that they had taken an OTC analgesic for a reason other than pain, most commonly to manage anxiety and aid sleep. University students also reported use for swelling/inflammation (*n* = 8), allergic reactions (*n* = 4), or as adjunct therapy in asthma (*n* = 1) or infection (*n* = 1). One university student self-reported paracetamol use in a suicide attempt. 

Having a parent that worked in health care did not appear to significantly influence safety perceptions or reports of non-indicated use in the sample of 68 young adults with parents in this sector.

## 4. Discussion

The Australian young adults surveyed used analgesics infrequently, generally appeared safety conscious, and only a small proportion reported taking OTC analgesics for an inappropriate indication. Analgesic use was informed by source of information, experience with and accessibility of the medicine, cost and other factors. Although health-related advice from doctors and pharmacists was highly valued, school-leavers prioritised parental guidance, whereas university students prioritised internet-based information as their key source of information on OTC analgesics.

Reports of infrequent analgesic use amongst the young adults surveyed contrasts with more frequent analgesic use by their international counterparts [6,11], and with findings that three-quarters of Australian adults, mainly aged 25 and older, reported using an analgesic in the previous month [2]. Higher use in older adults could reflect age-related pain or changing medication behaviours and international research included young people in hospital, possibly with other health conditions [11]. Whilst infrequent use of analgesics in young people is encouraging, this sample does not necessarily reflect the views of all young adult Australians. Comparative, longitudinal studies could confirm the findings of this study in a broader youth-based population and beyond.

Female gender and concurrent chronic health conditions were associated with more frequent use of analgesics. Our finding confirms the influence of gender reported in selected studies [23,46], yet contradicts others [11,20,26], and may actually reflect female management of menstrual pain [5,47], or the population recruited into this study. Alternatively, the societal construct around self-acknowledgement of pain and associated stigma has been postulated as a reason for this, with males not taking analgesics in order to ‘tough it out’ [5]. University students in this study indicated that they would ‘tough it out’ until pain became intolerable and this may reflect pain management strategies or it may reflect the influence of cost. Qualitative research could comparatively explore this construct further in young Australians. The influence of a concurrent chronic condition is consistent with studies by Albatti et al. and Denisova et al. [25,26] who also highlight that this population may have better medication knowledge and self-medication practices. Qualitative exploration could comparatively explore more frequent use of analgesics with the concept of toughing it out amongst a diverse sample of young adults to inform a strengths-based approach to education on effective pain management.

Many of the young adults surveyed acknowledged the potential risks associated with analgesic use and self-reported using analgesics for correct indications. However, safety is an ambiguous term, and this study did not explore interrelated notions of risk and safety in depth. Our young adults’ knowledge of analgesic safety could be attributed to Australian high school education which teaches adolescents how to access and research health information to inform their decisions from as young as 12 years old, although this is predominantly about harm-minimisation strategies for illicit drugs and alcohol consumption [48]. A smaller proportion of participants in our study believed analgesics to be completely safe and without harm, similar to young adults in Slovakia [17] and Poland [49]. Greater ambivalence over safety was noted in school-leavers, which may reflect more limited experience with analgesics, less autonomy or greater parental influence. Qualitative research could also explore parental perceptions of safety and the influence of these on ambivalence over safety and off-label use.

Whilst it is encouraging that most respondents correctly recognised that there are risks associated with the use of OTC analgesics, approximately one-quarter reported using OTC analgesics for a reason other than pain, most commonly anxiety or sleep. Off-label analgesic use for anxiety or sleep was also found in young adults in Sweden [5], Pakistan [50], and Poland [49], possibly reflecting similar misconceptions about the marketed indications. In Australian youth, using OTC analgesics for sleep may be partly explained by restriction of access to codeine-containing analgesics in February 2018 [51]. It is plausible that prior to this, young people potentially used codeine-containing products for their sedating effects but were unaware of which agent was sedating so now use simple analgesics instead. Using OTC analgesic for anxiety or sleep can introduce risk of harm if contraindicated in that person or through lack of access to more appropriate management strategies. A small number of health science students agreed that taking more medication than recommended on the pack would elicit a better effect. Whilst it is of concern that students enrolled in a health degree should be more informed, we did not identify specific field of study or progress through degree. It cannot always be assumed that health practitioners have good health literacy, and further research within this population is warranted [52]. Furthermore, medication-related decisions can be influenced by multiple other factors. Greater insight into how these factors interact is needed because if we were to extrapolate this off-label use to the broader population, nearly 750,000 of the three million young Australian adults aged between 15 and 24 years (2016) [53] could be exposed to risk of analgesic-related harm.

Cost, previous medication experience, accessibility and brand influenced young peoples’ choice of analgesic, with medication packaging and tablet appearance less important factors [5]. Analgesic cost was more important to university students, which may reflect independent living arrangements. Pharmacy as preferred location of purchase could support this finding, as larger pack sizes are available at lower prices than smaller pack sizes from supermarkets. Internationally, pack sizes were reduced to limit potential for analgesic-related harm [12]. The risks of larger analgesic pack sizes for Australian youth is unknown particularly if medications are inappropriately shared; University students in this study reported a higher rate of sharing OTC analgesics with friends than school-leavers. A systematic review reported higher prevalence of sharing prescription medication amongst young female adults, particularly if there was a similar ailment involved or a desire to help another person [54]. However, the studies in the review failed to investigate whether the appropriateness of the medication is assessed before sharing [54]; and further investigation is warranted.

A concept mostly unexplored in the literature was how young adults manage pain symptoms and how quick they are to self-medicate. School-leaver respondents were possibly more likely to use an analgesic at the first sign of pain if living at home, with increased accessibility to analgesics or influenced by parental guidance. Older adults use analgesic more frequently and parents, especially mothers, have been identified as important sources for teaching children about pain recognition [24,55]. However, in 2004, Australian mothers reportedly used paracetamol products as a ‘cure all’ for their children, believing it acted as a sedative, mood booster, and would calm a hyperactive child [56]. If misconceptions are communicated by parents or carers to children, this could potentially contribute to inappropriate OTC analgesic use and subsequent risk of harm. University students, typically attempted to initially treat the cause of the pain prior to self-medicating with OTC analgesics, e.g., drinking water for a headache associated with dehydration, possibly influenced by analgesic cost, prior experience or internet-based information. The credibility of the internet-based information was not assessed and the sustained impact of high-school education on appropriate sourcing of internet information is also unknown.

University respondents primarily preferred the internet as a source of analgesic information, similar to studies in Slovakia [17] and Russia [25], whereas school-leavers strongly preferred their parents, as observed in Norway [57] and Sweden [5]. Our findings could reflect maturation and developing autonomy in seeking information during enrolment in higher education, or other factors not yet explored. Encouragingly, both university students and school-leavers highly rated pharmacies as a trusted and preferred source of analgesic information, consistent with international literature [5,20,26]. However, it should be noted that whilst information about appropriate indications, safe dosing, and combining different analgesic classes can be obtained from pharmacies, the depth of information provided may vary depending on whether a consumer interacts with a pharmacist or a pharmacy assistant. Additionally, the aforementioned information is easily accessible on the product packages, yet adequate functional health literacy is required to comprehend and apply this information [20,33]. Further, information printed on product packages or through Consumer Medicines Information (CMI) leaflets may have low readability and may be difficult for consumers to interpret [58,59]. Young adults also appear to read medication labels less frequently compared to elderly consumers [21].

## 5. Strengths and Limitations

A key strength of this research was the insights collected from 302 young adults in two different life stages: leaving school and undertaking university education. The findings provide important initial insights into how young Australian adults perceive and use OTC analgesics and identify important areas for further research. Key insights reflect divergence in usage patterns and perceptions from international cohorts and older Australian adults. Study participants were primarily from one state within Australia and further investigation is warranted to confirm study findings and more fully explore underlying influential factors and changes in safety perceptions or analgesic use over time.

Study limitations include selection bias, social desirability bias through self-report and reliance on participant recall. Our findings may not be generalisable to young adults who reside outside of Queensland or who attend other universities within and outside of Queensland, do not finish school, or attend university. Selection bias towards health-based students may have occurred due to the nature of student enrolments in a non-compulsory summer trimester and participant views may not represent all university students. This, in particular compares to the school-leavers cohort which were recruited during one day at a Queensland-based school-leavers festival. Self-selection bias may have occurred in both groups towards those who have an interest in non-prescription medications, due to the voluntary nature of the survey. A response rate was unable to be calculated due to the multimodal recruitment strategies utilised. However, this ensured broader distribution of the survey than in-person single-campus recruitment. Participants were not asked knowledge-based questions and the potential influence of knowledge or health literacy cannot be determined. In addition, participants were not assessed or observed on how they would actually use analgesics; further research with a larger sample size is needed to confirm whether young adults are safety conscious in this context. A larger sample size would also be needed to confirm initial trends apparent in the data related to the potential influence of range of factors over safety perceptions related to analgesic use. However, reports of non-indicated use signify possible knowledge deficits which is to be further explored in an ongoing follow up study.

## 6. Conclusions

Australian young adults were using analgesics infrequently, and many acknowledged the potential risks associated with analgesic use. However, one-quarter of respondents self-reported OTC analgesic use for a non-indicated or off-label reason. With nearly 3 million young adults in Australia, it is important to further explore the factors underlying medication-related decisions, associated safety perceptions and misconceptions, and how these interact. This will facilitate identification of the risks of associated analgesic-related harm in young people and inform targeted campaigns to minimise this.

## Figures and Tables

**Figure 1 pharmacy-09-00054-f001:**
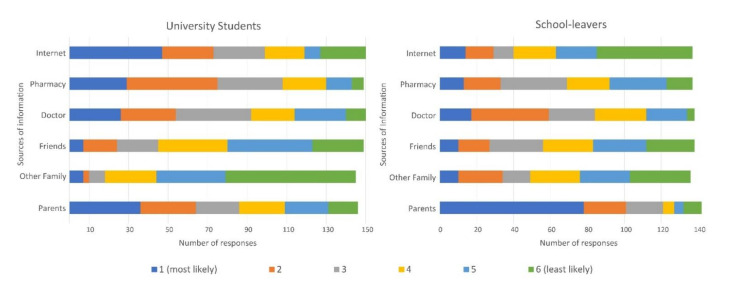
University students’ and School-leavers’ preference for source of OTC analgesic information.

**Table 1 pharmacy-09-00054-t001:** Gender, age and place of residence for participants.

Demographic Information	School-Leavers	University Students	Total
	(*n* = 147)	(*n* = 155)	(*n* = 302)
	*n* (%)	*n* (%)	*n* (%)
**Gender (*n* = 302) ^a^**			
Female	85 (57.8)	111 (71.6)	196 (64.9)
Male	61 (41.5)	41 (26.5)	102 (33.8)
Non-binary	0 (0)	2 (1.3)	2 (0.7)
Other ^b^	1 (0.7)	1 (0.6)	2 (0.7)
**Age (years) (*n* = 301) ^c^**			
17	121 (82.3)	3 (1.9)	124 (41.1)
18	24 (16.3)	18 (11.6)	42 (13.9)
19	2 (1.4)	29 (18.7)	31 (10.3)
20	0 (0)	27 (17.4)	27 (8.9)
21	0 (0)	23 (14.8)	23 (7.6)
22	0 (0)	16 (10.3)	16 (5.3)
23	0 (0)	12 (7.7)	12 (4.0)
24	0 (0)	11 (7.1)	11 (3.6)
25	0 (0)	15 (9.7)	15 (5.0)
**State or Territory (*n* = 302) ^d^**			
New South Wales	5 (3.4)	8 (5.2)	13 (4.3)
Victoria	0 (0)	3 (1.9)	3 (1.0)
Queensland	136 (92.5)	142 (91.6)	278 (92.)
Western Australia	6 (4.1)	2 (1.3)	8 (2.6)

^a^ Participants selected prefer not to answer for the following questions: Gender (*n* = 1), Aboriginal or Torres Strait Islander Person (*n* = 2), Chronic health condition (*n* = 3); ^b^ One school leaver indicated prefer not to answer and one University student self-identified as ‘N’; ^c^ Missing one response; ^d^ No responses from people living in South Australia, Tasmania, Australian Capital Territory or Northern Territory.

**Table 2 pharmacy-09-00054-t002:** Factors influencing OTC analgesic decision making in young adults.

Factors ^a^	School-Leavers	University Students	Total
	(*n* = 147)	(*n* = 155)	(*n* = 302)
	*n* (%)	*n* (%)	*n* (%)
Previous experience	65 (44.2)	115 (74.2)	180 (59.6)
Cost	43 (29.3)	104 (67.1)	147 (48.7)
Whatever I have access to	33 (22.4)	49 (31.6)	82 (27.2)
Brand	30 (20.4)	46 (29.7)	76 (25.2)
Medicine size/shape	14 (9.5)	38 (24.5)	52 (17.2)
Taste/coating	16 (10.9)	28 (18.1)	44 (14.6)
Medicine packaging	14 (9.5)	15 (9.7)	29 (9.6)
I do not take pain medicines	12 (8.2)	6 (3.9)	18 (6.0)
Other	1 (0.7)	15 (9.7)	16 (5.3)

^a^ Participants could select multiple factors, so data will not add to 100%.

**Table 3 pharmacy-09-00054-t003:** Responses to perception of safety.

Statement	Population Subset	Agree	Neutral	Disagree	I Don’t Know	Total
		*n* (%)	*n* (%)	*n* (%)	*n* (%)	*n* (%)
*Taking more medicine than stated on the pack gives a better effect*	University Students	24 (15.6)	14 (9.1)	101 (65.6)	15 (9.7)	154
	School-Leavers	13 (9.0)	22 (15.3)	93 (64.6)	16 (11.1)	144
	Total	37 (12.4)	36 (12.1)	194 (65.1)	31 (10.4)	298
*I think OTC pain medicines are safe (they cannot hurt me)*	University Students	46 (30.0)	24 (15.7)	80 (52.3)	3 (1.9)	153
	School-Leavers	45 (31.5)	41 (28.7)	45 (31.5)	12 (8.4)	143
	Total	91 (30.7)	65 (22.0)	125 (42.2)	15 (5.1)	296
*I have shared/would share pain medicines I can buy from the supermarket and OTC at the pharmacy with my friends*	University Students	121 (79.6)	11 (7.2)	19 (12.5)	1 (0.7)	152
	School-Leavers	51 (36.4)	36 (25.7)	42 (30.0)	11 (7.9)	140
	Total	172 (58.9)	47 (16.1)	61 (20.9)	12 (4.1)	292

## Data Availability

The data presented in this study are available on request from the corresponding author within ethics approval.

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
