# Peer review of "What Do Young Adults Think about the Safety of Over-the-Counter Analgesics? Findings from a Cross-Sectional Survey"

_pharmacy, 2021, doi:10.3390/pharmacy9010054_

Round 1

Reviewer 1 Report

Authors, thanks for doing this work!

Line 17: add a hyphen to school-leavers. This also applies to multiple spots throughout the paper, including lines 102, 105, 129, 186, 189, 210, 211, 248, 332.

Line 17: would a better term be "deferred" rather than "referred"?

Line 39: I did not check out the actual references, but hopefully they back up the notion that people might feel such medicines are "harmless". 

Line 55: extra space in the reference spacing.

Line 72: is there really peer pressure on young adults to take an analgesic? Can't really fathom that. Illicit drugs, of course. 

Line 110: sampling a school within the health field will obviously move the sample away from 'typical young adults'. You addressed some of this in Limitations, good.

Lines 113-127: a concern for a reader will be the lack of background used to create the 17 item tool. For example, there is lots of precedence for the items of Table 3. IF none of that literature was used to help create the items, okay fair enough, don't now say they were used. But if some were used, and just forgotten to be referenced, that all should get mentioned here. There are quite a few papers out there that attempted to come up with survey items on how to assess risk via analgesic use/OTC med use, for example. I do not expect you to add any items after-the-fact, but even with the pilot testing for content validity, you might want to buttress up this section if you can.

Line 144: I am not quite following the suggestion that '5-point Likert scales were recoded'. Are you saying "Strongly Agree" and "Agree" were collapsed to just "Agree"? Was that for stats purposes, to have less compartments to deal with?

Line 151-52: one decimal is lots, as you morphed to later on in the paper.

Line 167-175: I think it would be wise to list out the options for frequency of use (likely in Methods)? I see rare, infrequent, daily, monthly. Is that it? Did you use any definitions for subjects when selecting the first two?

Line 169: the statistical measure is not listed within the brackets. This was missed in other spots too (176-178, 213, 238-42). This stands out b/c the test was listed in a lot of other statistical statements.

Line 201-2 (later on Line 324): when stating a person 'wanted to treat the cause before', not really sure what that means. Do you mean with non-medical measures like a cold or hot pack?

Line 211: spacing at 72.4%

Line 213: another of the 2 key issues you may have (the first being lit for tool creation) is how the stats were done. On Line 213, it would appear GENDER was the grouping variable, then a host of options for a response. That would create Chi Sq tables along the lines of 2x5 or so. Those are not easy to do, I find. I assume your statistician signed-off on doing that approach and guided the analysis? The same applies to Lines 222-226 and 238-242 -- 2 categories paired to multiple response options (% each was chosen). 

Lines 272, 274, 283, 359: I wonder if you are using terms that are too strong for the results found -- "acknowledged potential risks", "judicious", "safety conscious"? You did comment on this a bit later, but even with that, you might want to soften up your stance. For instance, only 3 items were used for risks, and for that set, side effects were not even addressed. While many use the agents infrequently, we don't really know what doses are used at those times. On Line 171, 43 of 50 are using analgesics daily, yet little insight past that. So overall, I am a bit leery of the upbeat terms used. You might be right, but a bit hard to say with the minimal insight found during the surveys.

Line 292-293: an "IF" scenario is presented (if this happened, and that also was the case), but I wonder if you need some more insight to back-up the thinking here? Small point only, I can live with how you have it now.

Line 335: to play devil's advocate, all the information you list out is not only obtained from pharmacists, it is on every product package. Do you give too much credence for the necessity of pharm-consumer interaction here? Doing so might imply none of them read that (which I can believe for sure, but ...).

Line 347: I think you should give far more attention to selection bias here, just so the reader knows the different sources of subjects, and the style they were retrieved, gets due diligence in the write-up. 

Author Response

Authors, thanks for doing this work!

Thanks for reviewing it!

  1. Line 17: add a hyphen to school-leavers. This also applies to multiple spots throughout the paper, including lines 102, 105, 129, 186, 189, 210, 211, 248, 332.

Thankyou, a hyphen has been added to all uses of the term school-leavers for consistency throughout the paper.

  1. Line 17: would a better term be "deferred" rather than "referred"?

On reflection, we agreed that the term ‘deferred’ may better suit young adults utilising their parents as information sources for analgesics, rather than ‘referred’. This has been changed and is now reflected in line 18 in this copy.

  1. Line 39: I did not check out the actual references, but hopefully they back up the notion that people might feel such medicines are "harmless". 

The references have been checked and in once case the original reference number 6 has been substituted with an alternative reference that provides more relevant evidence to support this statement.

  1. Line 55: extra space in the reference spacing.

Corrected.

  1. Line 72: is there really peer pressure on young adults to take an analgesic? Can't really fathom that. Illicit drugs, of course. 

On review of the reference, it is apparent that this was more peer influence than peer pressure and the manuscript has been changed to reflect this.

  1. Line 110: sampling a school within the health field will obviously move the sample away from 'typical young adults'. You addressed some of this in Limitations, good.

Thank you for this feedback.

  1. Lines 113-127: a concern for a reader will be the lack of background used to create the 17-item tool. For example, there is lots of precedence for the items of Table 3. IF none of that literature was used to help create the items, okay fair enough, don't now say they were used. But if some were used, and just forgotten to be referenced, that all should get mentioned here. There are quite a few papers out there that attempted to come up with survey items on how to assess risk via analgesic use/OTC med use, for example. I do not expect you to add any items after-the-fact, but even with the pilot testing for content validity, you might want to buttress up this section if you can.

Thank you for addressing this; initially, the literature was used to develop the survey items and tracked in a different document. The following text has been added to section 2.2 lines 122 to 126:

“Questions were informed by the literature or adapted from published surveys. In the final survey, one question was adapted from each of four studies including Kelly et al. [18], Shone et al. [10], Holmström et al. [5], and Klimaszova et al. [16]. Although some of these were validated surveys, these instruments did not meet all aims of this particular study, and thus selected questions were adapted to address the specified objectives.”

  1. Line 144: I am not quite following the suggestion that '5-point Likert scales were recoded'. Are you saying "Strongly Agree" and "Agree" were collapsed to just "Agree"? Was that for stats purposes, to have less compartments to deal with?

Yes, that is correct that ‘strongly agree’ and ‘agree’ were collapsed to ‘agree’, and you are correct that this was for stats purposes, particularly when multiple factors were being compared. This information has been included in the ‘Data Analysis’ section (Section 2.3), on lines 154-160, as noted below:

“Selected variables, such as the five-point Likert-scale question including Agree Strongly, Agree Somewhat, Neutral, Disagree Somewhat, Disagree Strongly, or I don’t Know, were recoded to Agree, Disagree, Neutral, or I don’t know options. In addition, frequency of analgesic use was recorded from the original options: Never, Rarely, Few times a year, Every week, Every month or Every day, into Never/Rarely, Few times a year, Monthly, or Daily/Weekly. These selected variables were collapsed when multiple factors were included in analysis for statistical purposes.”

  1. Line 151-52: one decimal is lots, as you morphed to later on in the paper.

The manuscript has been reviewed and updated for consistency in this context.

  1. Line 167-175: I think it would be wise to list out the options for frequency of use (likely in Methods)? I see rare, infrequent, daily, monthly. Is that it? Did you use any definitions for subjects when selecting the first two?

This has been updated in the methods section of data analysis, please see response to point 8.

  1. Line 169: the statistical measure is not listed within the brackets. This was missed in other spots too (176-178, 213, 238-42). This stands out b/c the test was listed in a lot of other statistical statements.

The results section has been reviewed and amended for consistency as tracked changes on lines187, 195 and 196. In places the p value has been removed (lines 235 and lines 261-267) as a check of the analysis identified selected Chi-square tables with cells with a count of less than five, for example, in the 17 students that reported deferring to friends for advice. The text has been amended to reflect trends that could be investigated in future research (see point 14 for more information).

  1. Line 201-2 (later on Line 324): when stating a person 'wanted to treat the cause before', not really sure what that means. Do you mean with non-medical measures like a cold or hot pack?

Participants were asked to select their initial actions at first sign of pain from a list of options including:

‘I will use over-the-counter products like paracetamol (Panadol) or ibuprofen (Nurofen/Advil) at the first sign of pain’; ‘I try to treat the cause of the pain before self-medicating’; ‘I try to use other treatments like herbal products before using pain medicines’; ‘I don’t use pain medicines at all’; and an additional free text ‘other’ option.

In the instance of ‘trying to treat the cause of the pain before self-medicating’ the authors assumed that this could include a range of options, including non-pharmacological management such as hot/cold pack, or drinking water if dehydration causing headache etc.

The text in lines 222-223 has been revised to clarify that respondents selected an option.

The text in lines 353-354 has been amended as below.

“University students, typically attempted to initially treat the cause of the pain prior to self-medicating with OTC analgesics, e.g. drinking water for a headache associated with dehydration, possibly influenced by analgesic cost, prior experience or internet-based information.”

  1. Line 211: spacing at 72.4%

Corrected

  1. Line 213: another of the 2 key issues you may have (the first being lit for tool creation) is how the stats were done. On Line 213, it would appear GENDER was the grouping variable, then a host of options for a response. That would create Chi Sq tables along the lines of 2x5 or so. Those are not easy to do, I find. I assume your statistician signed-off on doing that approach and guided the analysis? The same applies to Lines 222-226 and 238-242 -- 2 categories paired to multiple response options (% each was chosen). 

A statistician was consulted prior to the study to advise on survey design and sample size and again as part of refinement of the data analyses plan, including use of simple inferential statistics such as Chi-square comparison of proportions in this sample. Likert scales were recoded to agree and disagree to allow for 2x2 comparison with factors such as gender and analyses has since been reviewed to ensure that cells with expected count of less than 5 did not exceed 20%. Where this is not the case, the text has been reviewed to indicate a similarity in trends and a statement has been added to the limitations to reflect this.  

See lines: 234-235, 261-267 and 395-397.

  1. Lines 272, 274, 283, 359: I wonder if you are using terms that are too strong for the results found -- "acknowledged potential risks", "judicious", "safety conscious"? You did comment on this a bit later, but even with that, you might want to soften up your stance. For instance, only 3 items were used for risks, and for that set, side effects were not even addressed. While many use the agents infrequently, we don't really know what doses are used at those times. On Line 171, 43 of 50 are using analgesics daily, yet little insight past that. So overall, I am a bit leery of the upbeat terms used. You might be right, but a bit hard to say with the minimal insight found during the surveys.

Thank you for your comment, you do raise an important point and it is difficult to state with certainty that young adults overall are making completely safe decisions. The text has been reviewed and the wording has been amended to soften the stance, e.g. removal of the word judicious. The limitations have been amended to also acknowledge that actual use was not assessed and findings relied on self-report.

Please see lines 297-302, 311-312, 393-397 and 401-403 for tracked changes.

  1. Line 292-293: an "IF" scenario is presented (if this happened, and that also was the case), but I wonder if you need some more insight to back-up the thinking here? Small point only, I can live with how you have it now.

On reflection, the ‘if’ scenario has been removed until more insight has been gathered.

  1. Line 335: to play devil's advocate, all the information you list out is not only obtained from pharmacists, it is on every product package. Do you give too much credence for the necessity of pharm-consumer interaction here? Doing so might imply none of them read that (which I can believe for sure, but ...).

Noted, the following text has been added to the manuscript in the discussion, on lines 367-372:

“Additionally, the aforementioned information is easily accessible on the product packages, yet adequate functional health literacy is required to comprehend and apply this information [19,32]. Further, information printed on product packages or through Consumer Medicines Information (CMI) leaflets may have low readability and may be difficult for consumers to interpret [57,58]. Young adults also appear to read medication labels less frequently compared to elderly consumers [20].”

  1. Line 347: I think you should give far more attention to selection bias here, just so the reader knows the different sources of subjects, and the style they were retrieved, gets due diligence in the write-up. 

Added on line 387-390:

“This, in particular compares to the school-leavers cohort which were recruited during one day at a Queensland-based school-leavers festival. Self-selection bias may have occurred in both groups towards those who have an interest in non-prescription medications, due to the voluntary nature of the survey.”         

Reviewer 2 Report

This is an interesting research which provides more information about use of analgesic drugs. However, I believe improvement can be done on the manuscript:

  1. line 11 - advice of health professional
  2. line 38-48 - this section should involve more data on adverse drug reactions of analgesics, for instance there is a publication for paracetamol serious adverse events reports from eudravigilance database

  3. line 148 - results section text should involve only percentages without n
  4. line 247 - Denisova et al.
  5. line 280 - excess spacing after leavers
  6. check excess spacing throughout the text

Author Response

Review Report Reviewer 2

This is an interesting research which provides more information about use of analgesic drugs. However, I believe improvement can be done on the manuscript:

  1. line 11 - advice of health professional

This has been amended on line 11.

  1. line 38-48 - this section should involve more data on adverse drug reactions of analgesics, for instance there is a publication for paracetamol serious adverse events reports from eudravigilance database

Thank you for this comment. We have referenced the Eudravigilance database publication on adverse events from paracetamol now in the paper, on line 47-49.

  1. line 148 - results section text should involve only percentages without n

This approach is currently considered good practice in our research discipline and we have left unamended. We seek the advice of the editor regarding any further change.

  1. line 247 - Denisova et al.

This has been corrected.

  1. line 280 - excess spacing after leavers

This has been corrected.

  1. check excess spacing throughout the text

The document has been checked for excess spacing and corrected accordingly.

Thank you to both of the reviewers for their comments, which have strengthened the

manuscript.
